# Monthly resolved modelled oceanic emissions of carbonyl sulfide and carbon disulfide for the period 2000-2019

Sinikka T. Lennartz[1], Michael Gauss[2], Marc von Hobe[3], Christa A. Marandino[4]

[1]Institute for Chemistry and Biology of the Marine Environment, University of Oldenburg, Carl-von-Ossietzky-Straße 9-11, 26129 Oldenburg, Germany
[2]Norwegian Meteorological Institute, PO 43 Blindern, 0313 Oslo, Norway
[3]Institute for Energy and Climate Research (IEK-7), Forschungszentrum Jülich GmbH, 52425 Jülich, Germany
[4]Geomar Helmholtz-Centre for Ocean Research Kiel, Düsternbrooker Weg 20, 24105 Kiel, Germany

*Correspondence to*: Sinikka T. Lennartz (sinikka.lennartz@uni-oldenburg.de)

**Abstract.** Carbonyl sulfide (OCS) is the most abundant, long-lived sulphur gas in the atmosphere and a major supplier of sulfur to the stratospheric sulfate aerosol layer. The short-lived gas carbon disulfide ($CS_2$) is oxidized to OCS and constitutes a major indirect source to the atmospheric OCS budget. The atmospheric budget of OCS is not well constrained due to a large missing source needed to compensate for substantial evidence that was provided for significantly higher sinks. Oceanic emissions are associated with major uncertainties. Here we provide a first, monthly resolved ocean emission inventory of both gases for the period 2000-2019 (available at https://doi.org/10.5281/zenodo.4297010)(Lennartz et al., 2020a). Emissions are calculated with a numerical box model (resolution 2.8° x 2.8° at equator, T42 grid) for the oceanic surface mixed layer, driven by ERA5 data from ECMWF and CDOM from Aqua-MODIS. We find that interannual variability in OCS emissions is smaller than seasonal variability, and is mainly driven by variations in chromophoric dissolved organic matter (CDOM), which influences both photochemical and light-independent production. A comparison with a global database of more than 2500 measurements reveals overall good agreement. Emissions of $CS_2$ constitute a larger sulfur source to the atmosphere than OCS, and equally show interannual variability connected to variability of CDOM. The emission estimate of $CS_2$ is associated with higher uncertainties, as process understanding of the marine cycling of $CS_2$ is incomplete. We encourage the use of the data provided here as input for atmospheric modelling studies to further assess the atmospheric OCS budget and the role of OCS in climate.

## 1 Introduction

The trace gases carbonyl sulfide (OCS) and carbon disulfide ($CS_2$) are naturally produced in the ocean and emitted to the atmosphere (Ferek and Andreae, 1983; Kettle et al., 2001; Khalil and Rasmussen, 1984; Watts, 2000). $CS_2$ is oxidized to a large extent to OCS (~82% on a molecular basis) within days after emission and thus constitutes a large indirect source in the atmospheric OCS budget (Chin and Davis, 1993; Stickel et al., 1993). OCS is the most abundant sulfur gas in the atmosphere with an average mixing ratio of ca. 480 ppt at land-based time series stations (Montzka et al., 2007) and ca. 550 ppt in the

marine boundary layer (Lennartz et al., 2020b). The sources and sinks of atmospheric OCS are important in two contexts: first, OCS is transported to the stratosphere due to its long tropospheric lifetime of 1.5 to 3 years (Montzka et al., 2007), where it is a major precursor of sulfate aerosols (Brühl et al., 2012; Kremser et al., 2016; Turco et al., 1980). The stratospheric sulfate aerosol layer influences the radiative budget by increasing the planetary albedo, and in addition provides surfaces for ozone

catalysing reactions (Solomon et al., 2011, 2015). Second, OCS has been suggested as a promising proxy to constrain the terrestrial $CO_2$ uptake on a global scale using inverse atmospheric modelling (Berry et al., 2013; Stimler et al., 2010; Whelan et al., 2018). In order to understand the dynamics of the sulfate aerosol layer and to apply OCS as a proxy for gross primary production, the quantification of OCS sources and sinks to the atmosphere on a global scale is required.

Currently, oceanic emissions are associated with the highest uncertainties among sources in the atmospheric OCS budget

(Kremser et al., 2016; Whelan et al., 2018). Evidence for increasing the vegetation sink led to a missing source the budget (Suntharalingam et al., 2008), and oceanic emissions have been suggested to account for a gap of 600-800 Gg S yr$^{-1}$ (Berry et al., 2013; Glatthor et al., 2015; Kuai et al., 2015b). Global oceanic emission estimates extrapolated from measurements range from -16 Gg S yr$^{-1}$ (Weiss et al., 1995b) to 320 Gg S yr$^{-1}$ (Rasmussen et al., 1982). Surface ocean models that are largely in agreement with observations report direct OCS emissions from the oceans of 41 Gg S yr$^{-1}$ (Kettle et al., 2002) to 130 Gg S yr$^{-1}$

$^{1}$ (Lennartz et al., 2017). Generally, surface seawater concentrations of OCS are too low to sustain emissions that would close the budget (Lennartz et al., 2017, 2020b). A detailed description of the marine emissions of OCS and its precursor $CS_2$ can serve as an input to modelling studies, and thus help to identify the missing source.

Models resolving the marine cycling of multiple trace gases are powerful tools to assess interannual variability of marine emissions through variations in the factors influencing production and consumption of the gas in seawater. The processes

determining OCS concentration in the surface ocean are better understood than those of $CS_2$, and model approaches for marine concentrations and emissions have been developed previously (Kettle, 2000; Kettle et al., 2002; Launois et al., 2015; Lennartz et al., 2017; Preiswerk and Najjar, 2000). While some show good agreement with observational data (Kettle et al., 2002; Lennartz et al., 2017; Preiswerk and Najjar, 2000), inconsistencies in calculating the hydrolysis rate (Lennartz, 2016) presumably led to overestimations in another study (Launois et al., 2015). All of these models use climatological forcing data.

For gases like OCS and $CS_2$ with a high spatiotemporal variability in their emissions, refining the temporal resolution of marine emission inventories would help to further constrain their atmospheric budget. Here we provide such a monthly resolved model output based on satellite data and reanalysis products.

The modelled processes include a photochemical production process, a light-independent dark production term, degradation by hydrolysis and air-sea exchange. Gas fluxes across the base of the mixed layer, i.e. diapycnal fluxes, seem to be of minor

importance, at least in tropical waters (Lennartz et al., 2019). The photochemical OCS production involves UV-radiation interactions with chromophoric dissolved organic matter (CDOM) (Ferek and Andreae, 1984; Modiri Gharehveran and Shah, 2018; Pos et al., 1998). Apparent quantum yields (AQY) decrease with increasing wavelength, but show orders of magnitude differences between locations (Cutter and Radford-Knoery, 1993; Weiss et al., 1995a; Zepp and Andreae, 1994). Reaction mechanisms involving thiyl radicals have been identified from precursor molecules such as cysteine, cystine and methionine

(Modiri Gharehveran and Shah, 2018; Pos et al., 1998). However, the complexity of the natural mixture of dissolved organic sulfur molecules in the ocean (Ksionzek et al., 2016) makes the determination of a photoproduction rate constant on a global scale difficult. Following an approach initially suggested by von Hobe et al., (2003), the photoproduction rate constant was scaled according to the CDOM absorption coefficient at 350 nm ($a_{350}$) in the global surface ocean box model used in this study (Lennartz et al., 2017). This approach led to good agreement of climatological mean modelled concentration with

measured sea surface OCS concentrations. The mechanism for OCS dark production is not well understood, and two not mutually exclusive hypotheses have been suggested, i.e. dark production being connected to abiotic radical reactions (von Hobe et al., 2001) or microbial remineralisation processes (Cutter et al., 2004). The dependency of the dark production rate on CDOM absorption and temperature shows good agreement across various biogeochemical regimes (Lennartz et al., 2019). Hydrolysis is the main chemical sink for OCS in the mixed layer. In both an acid and an alkaline reaction, OCS hydrolysis

yields $CO_2$ and sulfide (Elliott et al., 1987). This reaction is strongly temperature dependent, leading to e-folding lifetimes between several hours in warm waters and several days in cold, high latitude waters (Elliott et al., 1989). The temperature dependency of this reaction has been reasonably well described by independent laboratory and field studies (Cutter and Radford-Knoery, 1993; Elliott et al., 1989; Kamyshny et al., 2003).

$CS_2$ is present in seawater in picomolar concentrations, and measurements are generally sparse (Lennartz et al., 2020b). A

correlation between temperature and $CS_2$ concentration in surface waters is evident across several datasets (Lennartz et al., 2019; Xie and Moore, 1999). $CS_2$ is produced by photochemical reactions as well, following a similar shape of the AQY-wavelength spectrum as OCS (Xie et al., 1998). Precursor molecules such as cysteine, cystine, methionine and DMS have been identified, and photochemical $CS_2$ production itself seems to be temperature dependent (Modiri Gharehveran and Shah, 2018). Furthermore, there is evidence for a biological production of $CS_2$ by phytoplankton species, with varying yield from different

species (Xie et al., 1999), but the exact mechanism is unknown. Outgassing to the atmosphere is considered the most important sink process for $CS_2$ in the mixed layer. The only chemical sink mechanism known so far is hydrolysis with a lifetime of several years (Elliott, 1990). However, a chemical sink process in addition to air-sea gas exchange was needed to explain observations along an Atlantic transect, with an e-folding lifetime of ca. 10 days (Kettle et al., 2001).

Here, we use existing models that include parameterizations of processes known to be relevant for each gas, and apply them

on a global scale, accounting for interannual variability in the forcing parameters. We present the first monthly resolved inventory for marine OCS and $CS_2$ emissions for the period 2000-2019. The model is driven by diel cycles averaged over the course of each month or monthly averages of satellite data (Aqua-MODIS for CDOM) and ERA5 reanalysis products for meteorological parameters. We encourage the community to use these emissions for atmospheric modelling studies in order to elucidate the atmospheric budget of OCS, assess variability in the supply to the sulfate aerosol layer and determine gross

primary production on a global scale (available at: https://doi.org/10.5281/zenodo.4297010) (Lennartz et al., 2020a).

## 2 Model description

A model version as described in Lennartz et al. (2017) is used to model the interannual variability in oceanic emissions for OCS. A new model is developed to simulate oceanic emissions of $CS_2$. In both models, the surface ocean is divided into grid boxes of 2.8 x 2.8° at the equator (T42 grid, Gaussian grid with ~310 km resolution at equator (NCAR, 2017)) that comprise
various depth layers of 1m thickness depending on the depth of the mixed layer in each grid box. Note that the model does not resolve physical transport between the boxes (see Lennartz et al., 2017, for details).

The numerical model simulating OCS seawater concentration and air-sea exchange (positive for flux from ocean to atmosphere) includes the processes photochemical production, light-independent production (termed 'dark production'), degradation by hydrolysis and air-sea exchange across the sea surface. The process rates are calculated as depicted in Fig. 1
based on meteorological (global radiation, wind speed, skin temperature) and physicochemical data (salinity, seawater pH, CDOM absorption, and dry mole air fraction). The processes photochemical production $\frac{d[OCS]_{photo}}{dt}$, dark production $\frac{d[OCS]_{dark}}{dt}$, hydrolysis $\frac{d[OCS]_{hydrolysis}}{dt}$ and air-sea exchange $\frac{d[OCS]_{ase}}{dt}$ are calculated according to equation (1), all in $\left[\frac{pmol}{L \cdot s}\right]$ (Fig. 1):

$$\frac{d[OCS]}{dt} = +\frac{d[OCS]_{photo}}{dt} + \frac{d[OCS]_{dark}}{dt} - \frac{d[OCS]_{hydrolysis}}{dt} - \frac{d[OCS]_{ase}}{dt} \tag{1}$$


Photochemical production is calculated as the product of UV radiation $UV\left[\frac{W}{m^2} = \frac{J}{m^2 \cdot s}\right]$, the absorption coefficient of CDOM at 350 nm $a_{350}$ [m$^{-1}$], and the photoproduction rate constant $p$ integrated over the mixed layer depth (MLD) according to equation (2):


$$\frac{d[OCS]_{photo}}{dt} = \int_{-MLD}^{0} UV \cdot a_{350} \cdot p(a_{350})dz \tag{2}$$

The photochemical rate constant $p\left[\frac{J}{mol}\right]$ is scaled with $a_{350}$ [m$^{-1}$], following a rationale suggested by von Hobe et al., (2003), which reflects that $a_{350}$ can be regarded as a proxy for both photosensitizer and sulfur source across large spatial scales. The
linear dependence between $a_{350}$ and $p$ is calculated based on fits to observational data from three major ocean basins as described in Lennartz et al., (2017). This wavelength-integrated approach has been shown to reproduce both local measurements from several cruises as well as global OCS observations (von Hobe et al., 2003; Lennartz et al., 2017). UV radiation below the sea surface is calculated according to solar radiation, zenith angle and wind speed following von Hobe et al. (2003) as described in Lennartz et al. (2017). The light field in each 1 m depth layer is calculated by reducing the incoming
short-wave radiation depending on the local absorption coefficient $a_{350}$. Photochemical production is then computed for each

layer individually, followed by integration over the entire mixed layer. This integration inherently assumes a well-mixed surface layer.

Dark production is calculated according to Lennartz et al. (2019). This reaction rate is an update of the original formulation by von Hobe et al. (2001), resulting in a semi-empirical relationship based on observations from a wider spatial range of observation than the initial study. In this formulation, the dark production rate depends on temperature and $a_{350}$ [m$^{-1}$] (eq. 3):

$$\frac{d[OCS]_{dark}}{dt} = a_{350} \cdot 10^{-6} \cdot e^{\left(57.2 - \frac{16\,200}{SST}\right)} \tag{3}$$

OCS hydrolysis is determined according to Elliott et al. (1989) and depends on temperature (T), salinity (SSS) and the proton activity $a[H^+]$ [-], equivalent to $10^{-pH}$, according to eq. (4) and eq. (5):

$$\frac{d[OCS]_{hydrolysis}}{dt} = [OCS] \cdot \left[\exp\left(24.3 - \frac{10\,459}{T}\right) + exp\left(22.8 - \frac{6\,040}{T}\right) \cdot \frac{K}{a[H^+]}\right] \tag{4}$$

$$-log_{10}K = \frac{3046.7}{SST} + 3.7685 + 0.0035486 \cdot \sqrt{SSS} \tag{5}$$

Air-sea exchange is calculated as the product of the concentration gradient between water and equilibrium concentration $\Delta c$ and the transfer velocity $k \left[\frac{m}{s}\right]$ parametrized according to Nightingale et al. (2000):

$$\frac{d[OCS]_{ase}}{dt} = k \cdot \Delta c \tag{6}$$

The equilibrium concentration is calculated according to de Bruyn et al. (De Bruyn et al., 1995) based on the atmospheric dry mole fraction where here, a fixed value is assumed (Tab. 1). The transfer velocity is corrected for OCS with the Schmidt number, calculated based on the molar volume according to Hayduk and Laudie (1974).

The model for CS$_2$ includes the processes of photochemical production and a first order chemical sink $\left[\frac{pmol}{L \cdot s}\right]$, according to eq. (7).

$$\frac{d[CS_2]}{dt} = +\frac{d[CS_2]_{photo}}{dt} - \frac{d[CS_2]_{chem.sink}}{dt} + \frac{d[CS_2]_{ase}}{dt} \tag{7}$$

Photochemical production is calculated in the same way as for OCS, with an additional reduction factor r [-] applied (eq. 8).

$$\frac{d[CS_2]_{photo}}{dt} = r \cdot \int_{-MLD}^{0} UV \cdot a_{350} \cdot p(a_{350})dz \tag{8}$$

Xie et al. (1998) approximated that CS$_2$ photoproduction rates are about a factor of five smaller than OCS photoproduction rates by comparing an experimentally derived AQY from CS$_2$ and OCS (r=0.2 in eq. 8). The two AQY were not measured at the same location, but in comparable water properties. Another study with simultaneous measurements of both gases reported varying factors between 0.2 and 0.014 (5 to 70 times smaller than OCS photoproduction) (Lennartz et al., 2019). Here, we scaled the reduction factor to obtain the best fit in the average concentration, resulting in a factor r=0.1 in eq. 8. Thus, the model reflects the similar shape of the AQY for both gases by assuming a constant ratio, but the scaling of the overall magnitude of the photoproduction rate constant is chosen to obtain the best fit to observations from the database in (Lennartz

et al., 2020c). A chemical sink according to the model formulation in Kettle (2000), i.e. with an e-folding lifetime of 10 days $\left(\frac{1}{k_{cs}}\right)$, was implemented according to eq. (9), $k_{cs}$ in unit $\left[\frac{1}{s}\right]$

$$\frac{d[CS_2]_{chem.sink}}{dt} = k_{cs} \cdot (CS_2) \tag{9}$$

Air-sea exchange was calculated as described for OCS, using the $CS_2$ solubility according to De Bruyn et al. (1995).

As $CS_2$ cycling in the water column is not yet well understood, this model should be understood as a base model to be extended as soon as additional process rates and their dependencies become available.

## 3 Simulation set-up

Simulations are performed for the period 2000-2019. There are several changes in the forcing data compared to the climatological run in Lennartz et al. (2017). Here we use monthly resolved data for the period 2000-2019 for $a_{350}$, surface

shortwave radiation, surface (skin) temperature, wind speed and sea level pressure (Table 1). Skin temperature (diagnosed close to air-sea interface) is used as forcing data for all temperature-relevant processes, i.e. air-sea exchange, but also dark production and hydrolysis. To test the sensitivity of emissions on the choice between skin and sea surface temperature, we performed a sensitivity test for the year 2000. The meteorological data were obtained from the ERA5 reanalysis (more specifically, its product line 'ERA5 hourly data on single levels from 1979 to present', (Hersbach et al., 2018)) through the

Copernicus Climate Change Service (https://climate.copernicus.eu/). One file per year and parameter, containing hourly data on 0.25° x 0.25° resolution, was downloaded.

For wind speed, the zonal and meridional components of wind speed at 10m altitude $\left[\frac{m}{s}\right]$ ($u_{10}$ and $v_{10}$, respectively) were downloaded separately and converted into wind speed *ws* according to

$$ws = \sqrt{u_{10}^2 + v_{10}^2}$$


The post-processing of the meteorological data was done using CDO tools (climate data operators, version 1.9.8) (Schulzweida, 2019) and comprised the following steps:

a) the yearly files for each parameter were split into monthly files using the CDO flag 'splityearmon', resulting in 240 monthly files covering the 20-year period 2000 to 2019 for each parameter;

b) for each of the 240 months within the period, monthly-mean diel cycles of each meteorological parameter *x* were calculated using the CDO flag 'dhouravg', which calculates multi-day averages for every hour of a day as

$$\bar{x}_m(h) = \frac{1}{N_m} \sum_{d=1}^{N_m} x(d,h)$$

where *m* is the month (1 to 12), *h* is the hour of the day (1 to 24), *d* is the day of the month (1 to 28, 29, 30, or 31), and $N_m$ is the number of days within month *m*;

c) the resulting fields were regridded from the regular 0.25° x 0.25° longitude-latitude grid into the spectral T42 grid (~2.8° x 2.8°) using the cdo flag 'remapcon2', which is a second-order conservative remapping method that takes into account all source grid points, both in longitude and latitude directions. The spatial resolution is the same as in Lennartz et al., (2017). Among the remapping methods available in CDO, 'remapcon2' was considered the most appropriate to interpolate the selected meteorological parameters from a fine grid to a much coarser grid. Monthly forcing fields for CDOM are derived from Aqua MODIS satellite level 3 product 'absoprtion due to gelbstof and detritus at 443 nm' (NASA Goddard Space Flight Center, 2019), and converted to 350 nm with an exponential slope of 0.02 for the wavelength spectrum. Climatological values are used for salinity and mixed layer depth at a monthly resolution, which is the same for each month of the year throughout the simulation period, unchanged compared to Lennartz et al. (2017). The average diel cycle of each meteorological dataset (wind, pressure, skin temperature and solar radiation) is used for the 15th of each month (one value for every 2 hours).. In between, data is interpolated separately for each time of the day, resulting in a continuous change of the amplitude of the diel cycles. This procedure avoids sharp changes as if a mean monthly cycle was used for each day of the months, while still being computationally effective. The initial concentration for both gases was taken as a constant value of 8 pmol $L^{-1}$ in all grid boxes. The time step in the model is 2 hours. The model is spun up for one year, repeating the conditions of year 2000 prior to the simulation period. Maps were created using the m_map package v1.4m (Pawlowicz, 2020).

## 4 Results

### 4.1 Spatial and seasonal variability

Both gases show distinct spatial patterns in their annual concentration and emission averages, which reflect their marine cycling. For OCS, highest concentrations are present in cold, high latitude waters and shelf areas, whereas lowest concentrations prevail in warm, subtropical gyres where CDOM abundance in the water is low (Fig 2a). A latitudinal gradient with higher concentrations in high latitudes and low concentrations in tropical and subtropical waters reflects the temperature-dependent degradation by hydrolysis. The degradation is strongest in warm waters, where the lifetime of OCS is on the order of hours, keeping concentrations low. This general pattern is in broad agreement with observations of the largest available database on seaborne OCS measurements (Lennartz et al., 2020c). Annual mean emissions largely follow the spatial pattern of OCS sea water concentrations, with sources, i.e. flux from the ocean to the atmosphere, in shelf areas and high latitudes, and sink regions in the subtropical gyres (Fig. 2b). This general source and sink pattern does not change in all years covered in this period, but the absolute concentrations and, hence, the magnitude of the emissions, show variability (see Section 4.2). The concentration pattern follows the seasonal pattern of radiation that drives photochemical production, resulting in an annual cycle with highest concentrations and emissions in temperate northern latitudes in boreal summer and highest concentrations and emissions in the Southern Ocean in austral summer. The globally integrated monthly emissions are highest in austral summer and lowest in austral winter, due to the high emissions in the Southern Ocean, which outweighs the northern hemispheric summer emissions due to its large surface area, high wind speeds and high OCS seawater concentrations. The

amplitude of the mean seasonal cycle of OCS emissions is 21 Gg S yr$^{-1}$ (Fig 3b). In July and August, the globally integrated net emissions are close to zero, similar to a previous budget using a similar model (Kettle et al., 2002).

CS$_2$ concentrations show a different global pattern than OCS concentrations. CS$_2$ concentrations and emissions have hot spots in coastal and shelf regions, as well as in tropical and subtropical oceans, reflecting photoproduction as the main production process in the model. The tropical and subtropical areas show comparably low CS$_2$ concentrations (Fig. 4c), and their importance for globally averaged emissions mainly comes from the large oceanic surface area (Fig. 4d). Notably, CS$_2$ emissions in the western Pacific, where inverse modelling studies have located the missing OCS source, are relatively low

(Glatthor et al., 2015; Kuai et al., 2015b). The hot spots being located in the tropical and subtropical regions with similar intensities of incoming radiation all year, leads to less seasonal variation in globally integrated emissions, i.e. an amplitude of 3.2 Gg S yr$^{-1}$. The ocean is a source of CS$_2$ to the atmosphere over the entire year, since emissions are calculated with an atmospheric mixing ratio of 0 ppt. This assumption is a simplification, the average of the sparse dataset (less than a thousand measurements) on CS$_2$ air mixing ratios being 42±24 ppt, but ranging to not detectable in remote ocean regions. The difference

can be up to 30% in the computed flux, similar to the uncertainty inherent to the computation of the transfer velocity. In general, highest emissions occur in boreal winter, and the lowest in boreal summer.

## 4.2 Interannual variability

Surface concentrations of OCS show a similar spatial pattern across the period of 2000 to 2019, with interannual variability in

the absolute concentration and, hence, emissions. Globally integrated emissions range from 77.3 Gg S yr$^{-1}$ in 2001 to 142.1 Gg S yr$^{-1}$ in 2017, with a mean of 110.3±20.3 Gg S yr$^{-1}$ (Tab. 2). A significant increasing trend (p=0.028) is present in oceanic emissions from the period 2003-2019 of about 1.7 g S yr$^{-1}$ increase per year (Tab. 2). This trend is present also in the area-weighted average sea surface concentration (slope=0.007 pmol L$^{-1}$ yr$^{-1}$, p=8x10$^{-33}$). Note that for the trend analysis, we considered only the period 2003-2019, as CDOM seems to be one of the most important drivers of interannual variability (see

below), and CDOM data are only available from 2003 onwards. Generally, the seasonal variability of OCS emissions is larger (range of mean annual cycle of 21 Gg S yr$^{-1}$) than the interannual variability (mean monthly variability of 8.4 Gg S month$^{-1}$) (Fig. 3). Interannual variability of the emissions in each month is largest during boreal spring (April, May, June) and fall (October) (Fig. 3a). These months show the largest difference between minima and maxima during the whole period (grey area in Fig 3a). The spatial pattern of interannual variability of OCS emissions shows highest variability, i.e. highest standard

deviation among annual averages in each gridbox, at locations with high OCS concentrations and emissions (Fig. 2). These regions comprise the northern temperate and polar regions, the Southern Ocean, and shelf areas, especially those close to coastal upwelling regions and river plumes (Fig. 2). The standard deviation for OCS concentrations between annual averages ranges from 0.22 at the oligotrophic gyres to 143.8 pmol L$^{-1}$ at the highly dynamic coast off Alaska, USA (average standard deviation 3.4 pmol L$^{-1}$). The interannual variability also shows latitudinal differences. Polar regions in both Arctic and

Antarctic waters display the largest seasonal cycles in OCS concentration, i.e. the highest annual variability (Fig 4), and at the same time also display highest interannual variability. Differences in mean concentrations (area weighted) in summer range between 72.8 pmol L$^{-1}$ in June 2011 to 91.6 pmol L$^{-1}$ in July 2017, i.e. ca. 20 pmol L$^{-1}$ in the Arctic ocean (Fig. 4). Interannual differences in mean monthly OCS concentrations become smaller with decreasing latitudes, and are lowest in tropical oceans where they range between 7.0 pmol L$^{-1}$ in April 2002 and 8.5 pmol L$^{-1}$ in April 2018 (south tropical) and 8.6 pmol L$^{-1}$ in June

2015 and 9.0 pmol L$^{-1}$ in June 2018 (north tropical). Due to their large surface area and medium surface OCS concentrations, southern temperate regions (23°S-66°S) have the largest integrated OCS emissions, followed by northern temperate regions (33°N-66°N) (Fig. 4). In temperate regions, largest interannual variability occurs during the months of maximum positive emissions, with a range from 17.4 to 26.1 Gg S month$^{-1}$ in southern temperate regions in December, and 14.0 and 20.9 Gg S month$^{-1}$ in northern temperate regions in May. In summary, OCS concentrations and emissions show the highest interannual

variability at time and locations where concentrations are high, and in systems that are inherently highly dynamic such as shelf regions.

Carbon disulfide concentrations are highest in shelf areas in the tropics and subtropics, and generally decrease towards high latitudes (Fig. 2c). The spatial pattern of the annually integrated emissions mirrors this picture (Fig. 2d). While the spatial

pattern of concentrations and emissions is similar in each year, the absolute concentration and magnitude of emissions does show interannual variability (Fig. 3b). Emissions are calculated here with a boundary layer mixing ratio of zero (maximum possible emission) as is commonly done for other short-lived gases such as DMS (Lana et al., 2011), so the ocean is a CS$_2$ source at every location throughout the year. Globally integrated emissions range from 160.0 Gg S yr$^{-1}$ in 2002 to 189.7 Gg S yr$^{-1}$ in 2017 (Tab. 2). Similar to OCS, an increasing trend of global CS$_2$ emissions for the period 2003-2019 is significant

(p=0.0067). Emissions increase with 0.95 Gg S per year on average over the period 2003-2019. For globally integrated emissions, annual variability (mean range of 3.2 Gg S month$^{-1}$) is comparable to the interannual variability (3.2 Gg S yr$^{-1}$). This is different to OCS, where annual variability was higher than interannual variability for globally integrated emissions. This difference is caused by the location of the respective hotspots of the produced gases: As OCS has its concentration and emission hot spots mainly in high latitudes, which experience a very seasonal light regime, its annual variability is high. The

low concentrations of OCS (and corresponding low emissions) in the tropics result from the fast degradation by hydrolysis. In contrast, CS$_2$ has its concentration and emission hotspots mainly in low latitudes with more constant forcing, and hence displays smaller annual variability. The interannual variability of CS$_2$ emissions among single months has a similar magnitude throughout the year (grey shaded area in Fig. 3d). Maximum monthly mean concentrations of CS$_2$ vary the most in the summer months of the northern temperate regions (23°-66°N) from 4.3 Gg S month$^{-1}$ in June 2011 and 6.0 Gg S month$^{-1}$ in June 2018,

but show less variability in the winter months, i.e. between 0.8 and 1.2 Gg S month$^{-1}$ in December. Due to their comparably low surface area and the relatively low concentrations, high latitude regions do not play a significant role in globally integrated CS$_2$ emissions (Fig. 4). The dominance of southern temperate emissions of CS$_2$, despite higher absolute mean concentrations in northern temperate regions is explained by the larger surface ocean area in the southern temperate regions (Fig. 4c and d).

### 4.3 Main drivers of interannual variability

The interannual variability in OCS and $CS_2$ concentrations and emissions is a result of the interannual variability in their production and consumption processes, which in turn depends on environmental conditions. The variability comprises years like 2015 or 2017, in which positive OCS emissions occur in every month of the year, and years like 2019, where global net uptake by the ocean is present in four of the twelve months (Fig. 3a). Most of the interannual variability in these emissions are driven by the emissions in the high latitudes. For example, in 2017, emissions in the Arctic regions are higher than average, and lead to an overall increase in the emissions even in the winter months. 2015/2016 was a strong El Nino year, and decreased upwelling of cold water with high CDOM content would expectably lead to low OCS emissions due to decreased photochemical and dark production, and increased hydrolysis due to warmer water temperatures. However, as fluxes in the tropics are generally small, the global emissions are not substantially lower compared to other years (for 2015 they are even higher, due to higher emissions in high latitudes). The many negative fluxes in 2019 seem to result from lower than average emissions in the Southern Ocean.

Globally integrated annual emissions of OCS correlate significantly with global annual averages (area-weighted) of CDOM $a_{350}$, skin temperature and wind speed (Tab. 3). CDOM $a_{350}$ explains the largest variance, and sea surface temperature and wind speed explain less of the observed variance. Thus, CDOM $a_{350}$ has the strongest influence on the variability of global scale OCS concentrations. The influence is not surprising, as CDOM $a_{350}$ impacts both photochemical and dark production of OCS and modulates the light field in the water (at higher $a_{350}$, photoproduction is higher but also more limited to the surface). The photochemical production rate is second order dependent on CDOM $a_{350}$, reflecting its double role as photosensitizer, i.e. those molecules absorbing light energy for photochemical reactions, and as a proxy for the amount of sulfur molecules able to form radicals in photochemical reactions. As such, CDOM $a_{350}$ exerts a strong, non-linear and positive influence on OCS concentration, and seems to be the main driver of its interannual variability. The overall strong influence of CDOM $a_{350}$ on OCS interannual variability is also underlined by similarity in the spatial pattern of the standard deviation in annual average concentrations and emissions between OCS and CDOM $a_{350}$ (Fig. 5). Sea surface temperature strongly influences OCS hydrolysis, which leads to low concentrations in warm tropical and subtropical waters. Temperature also controls the solubility of the gas in water, i.e. the equilibrium water concentration is higher in colder waters. Variations in temperature explain a small part of interannual variations in OCS emissions. However, rising temperature towards the end of the period (Fig. 5) did not outweigh the increase in CDOM $a_{350}$, which supports the above mentioned result that the observed changes in CDOM $a_{350}$ had a stronger influence on overall OCS production than observed temperature changes had on hydrolysis. Finally, wind speed imposes a nonlinear control on OCS emissions, but the impact is smaller than that of CDOM $a_{350}$.

Resolving the correlations regionally shows distinct controls on interannual variability for CDOM and wind speed, but not for temperature (Fig. 6). Highest Pearson's correlation coefficients ($R^2$) for CDOM and OCS emission are found globally except for the subtropical gyres (Fig. 6a). In those gyre regions, CDOM concentration is generally low (Fig. 5a), so that other drivers

like wind speed seem to have a higher impact on the variability (Fig. 6e). Correlations with temperature show no clear spatial pattern (Fig. 6c).

Globally integrated $CS_2$ emissions correlate significantly with CDOM $a_{350}$, with a substantial part of the variance in interannual variability (67%) explained by this single factor, although this is less than for OCS. Photochemical production of $CS_2$ is similarly calculated as that for OCS, and hence depends nonlinearly and positively on CDOM $a_{350}$. The lesser amount of explained variance compared to OCS may result from the lack of a CDOM $a_{350}$ dependent dark production process. Interestingly, $CS_2$ emissions correlate with temperature, although temperature is not part of any production or consumption
process in the model, and solely modulates the solubility of $CS_2$. Increasing temperature decreases the solubility and would lead to a lower surface water concentration, hence, this effect cannot explain the correlation between temperature and $CS_2$ surface ocean concentrations in observations (Lennartz et al., 2020). Potentially, the co-variation of temperature with radiation dose might be responsible for the correlation of $CS_2$ concentration and temperature that is evident across observational datasets (see Introduction). The spatial variation of the standard deviation of annual averages of $CS_2$ concentration and emissions
resembles that of CDOM $a_{350}$, again underlining that this is a major factor for interannual variability of $CS_2$ (Fig. 5). Regional analysis of correlations of $CS_2$ emissions with biogeochemical and meteorological data shows that CDOM is a globally homogeneous driver of emissions as indicated by the high Pearson's correlation coefficients globally. Temperature and wind speed show highest correlation to $CS_2$ emissions in the tropical West Pacific, where the assumed source region of the 'missing source' of OCS is located. In these regions, interannual variability of wind speed is highest (Fig 5), and temperature shows
increased variability there (Fig. 5). This increased variability might explain the regionally strong correlation with $CS_2$ emissions.

### 4.4 Comparison to observations

    The model output of the monthly resolved simulation for 2000-2019 is compared to the database compiled by Lennartz et al.,
(2020), which contains 2970 fully georeferenced OCS measurements and 501 fully georeferenced $CS_2$ measurements in the period considered here. The model output is subsampled at time (including time of day) and location closest to the measurements in the respective period for a 1:1 comparison.

    For OCS, the range of the subsampled model output agrees well with data from the database (7 cruises, n=2971), with a slight underestimation of measured concentrations by the model (average 40.1 pmol $L^{-1}$ in the database, 38.4 pmol $L^{-1}$ in the model,
Fig. 7a). The direct comparison reveals remaining scatter around the 1:1 line, and a high bias in the model which grows with increasing OCS concentrations (Fig. 5b). A correction for this bias was obtained from a linear fit through the 1:1 comparison (blue dots in Fig 7), and yields the equation [OCS corrected] = 0.83 x [OCS modelled] – 0.7. Because the bias is still within the scatter of the data, we did not apply this correction factor in the analysis presented here. The scatter and high bias in the data likely results from simplifications in the model. The main simplifications, probably causing these discrepancies between

observation and models, are the missing horizontal transport, the use of averaged wind speed as forcing, the use of CDOM $a_{350}$ as a proxy for photochemical production and the application of a climatological mean for the depth of the mixed layer.

Using CDOM $a_{350}$ as a proxy for OCS photochemical production may introduce some scatter, but likely not a systematic bias. The very complex nature of the dissolved organic matter pool in the ocean, which comprises CDOM as the optically active fraction, makes it difficult to assign one photoproduction rate constant or apparent quantum yield to all the reactions taking

place with different precursors. CDOM $a_{350}$ has been shown to be a suitable proxy across three major ocean basins (Atlantic, Pacific, Indian Ocean), but the rate constant - CDOM $a_{350}$ relationship showed some scatter that might be improved when more data becomes available.

The missing horizontal transport can lead to a systematic model bias especially in cold waters where the OCS lifetime increases to time scales (days) relevant for physical transport while environmental conditions might vary on shorter time scales, but still

this process is unlikely to decouple OCS concentrations from its drivers like CDOM and temperature, that would be transported accordingly. Due to the short OCS lifetime in water, the effect of horizontal transport is negligible in warm waters of the tropics, subtropics and most of the temperate regions. In regions with deep mixed layers such as the Southern Ocean, the assumption of a completely well mixed surface layer may be violated and cause discrepancies between the modelled value (average of mixed layer) and the measured value (close to surface, i.e. higher concentration that at bottom of the mixed layer).

Since the modelled concentration depends on depth of the mixed layer and its relation to the photic zone, a climatological average as used here will introduce biases, however, detailed information on mixed layer depth in monthly resolution from observations is not available. This simplification mainly affects OCS concentrations in high latitudes, where concentrations are relatively high, and thus might be partly responsible for the systematic bias revealed by the scatter plot in Fig. 5b. Furthermore, averaging wind speed to a mean monthly cycle will most likely lead to an underestimation of emissions and,

hence, an overestimation of concentrations. Due to the nonlinear relationship of the transfer velocity of the gas exchange with wind speed, averaging disproportionally reduces the effect of increased emissions during high wind speeds. Another source of uncertainty is the forcing data, e.g. the choice of using the skin temperature rather than the sea surface data. For comparison, we performed a shorter simulation covering the year 2000 and using the ERA5 sea surface temperature data instead of the skin temperature. The difference in resulting global emissions was 1.2%, i.e. very small compared to other uncertainties. Still, given

these simplifications and assumptions, the overall good agreement with the measurements underlines the applicability of the model for assessing the marine cycling of OCS and its emissions to the atmosphere.

The marine cycling of $CS_2$ is less well understood than that of OCS. This relatively poorer process understanding is reflected by the comparison of the modelled $CS_2$ concentrations with those of the database (3 cruises, 501 measurements) ($R^2=0.04$).

Modelled concentrations agree with observations on average (average database: 18.0 pmol $L^{-1}$, average subsampled model output: 18.2 pmol $L^{-1}$). The three cruises cover the Mauritanean upwelling (Poseidon 269, blue in Fig. 7d), the Peruvian upwelling (ASTRA-OMZ, yellow in Fig. 7d) and a transect through the Atlantic (Transpegaso, green in Fig. 7d). As such, they cover a broad range of different biogeochemical regimes, but regions such as oligotrophic gyres or high latitude waters are not

covered, i.e. a substantial part of the global variability might be missing in the reference dataset. While the cruises Poseidon 269 and ASTRA-OMZ are relatively well represented by the model (colour code in Fig. 7d), the variability of the measurements during Transpegaso is not well captured. The model used here has some underlying assumptions and simplifications that call for refinement in the future when detailed process understanding is available. For example, the model is based on the assumption of a constant ratio between the apparent quantum yields of OCS and $CS_2$. It has been shown that this ratio is not always constant ((Kettle, 2000; Lennartz et al., 2019), but as the production pathways of both gases show some similarities (Modiri Gharehveran and Shah, 2018), the model formulation with a constant ratio is a first approximation. Second, the presence of a chemical sink is rationalised by its necessity to explain observed concentrations along an Atlantic transect (Kettle, 2000; Lennartz et al., 2019), but has no mechanistic foundation so far. Dedicated laboratory experiments disentangling the source and sink processes in the water column are needed to further resolve this issue and to improve modelling efforts. Finally, this model does not consider any biological production of $CS_2$. This assumption is justified for a first approximation, as CDOM and primary production (photosynthesis) show similar global scale patterns. High CDOM will thus lead to high production of $CS_2$ in the water, even though the scaling of the photoproduction rate constant (AQY) might inherently include biological production due to the covariation of photosynthesis patterns with CDOM and radiation. The calculated $CS_2$ emission estimate is not sensitive towards the choice of the temperature forcing data, resulting differences in global emissions when using the sea surface temperature instead of the skin temperature for the year 2000 resulted in a negligible deviation of 0.12%. Overall, the presented $CS_2$ concentration and emissions are a first approximation, and more detailed process understanding is important to improve emission estimates. Assuming that the presented oceanic emissions are in a realistic range, the calculated emissions would not be enough to close the gap in the atmospheric budget of OCS on the order of 600 to 800 Tg S yr$^{-1}$ (Berry et al., 2013; Glatthor et al., 2015; Kuai et al., 2015a), given that only a little more than half of the sulphur in $CS_2$ is converted to sulphur in OCS.

The emission estimate of the gases OCS and $CS_2$ includes further uncertainties introduced by the parameterizations of the transfer velocity used for calculation of air-sea exchange, which carry large uncertainties especially at high wind speeds (Wanninkhof, 2014). Furthermore, emissions here are calculated based on the concentration gradient between surface water and the equilibrium concentration dictated by the atmospheric mixing ratio, without taking into account any potential effect of the sea surface microlayer. Whether and how the enrichment of surfactants in the sea surface microlayer affects emissions of these gases has not been sufficiently assessed to date.

## 5 Data and code availability

The code is available on github under https://github.com/Sinikka-L/OCS_CS2_boxmodel. The simulation output is available at zenodo 10.5281/zenodo.4297010 (Link: https://doi.org/10.5281/zenodo.4297010)(Lennartz et al., 2020a). The output

consists of one netCDF files for each gas, each of a size of ca. 444 MB with monthly averages of sea surface concentrations and emissions to the atmosphere, as well as a mean diel cycle for each month.

## 6 Summary and Conclusions

OCS and $CS_2$ are climate relevant trace gases and OCS can also be used as a proxy to infer terrestrial gross primary production. A missing source in the atmospheric OCS budget currently makes conclusions on the future impact on both gases and the
application of this proxy on a global scale difficult. Since both gases contribute to the atmospheric OCS budget, their oceanic emissions have been suggested previously to account for that missing source. We provide monthly resolved OCS and $CS_2$ concentration and marine emission data for the period 2000-2019 based on a mechanistic ocean box model. We show that interannual variability of OCS is smaller than its seasonal variability in globally integrated emissions, but that a significant positive trend is evident across the period 2000-2019. The main driver for interannual variabilities is variation in CDOM $a_{350}$.
The comparison of our data to a database with more than 2500 measurements reveals an overall good agreement. The $CS_2$ model presented here for the first time is a first approximation and reveals stronger interannual variability than seasonal variability of emissions. Again, CDOM (or indirectly, biological production) seems to be strongly influencing concentration and emission patterns of $CS_2$. Similarly, an increasing trend in $CS_2$ emissions is significant for the period 2000-2019. Based on the data presented here, it seems unlikely that the missing atmospheric source of 600-800 Gg S $yr^{-1}$ (Berry et al., 2013;
Glatthor et al., 2015; Kuai et al., 2015a) might be balanced by tropical marine emissions of OCS or $CS_2$. We encourage the use of the data provided here as input for atmospheric modelling studies to further assess the atmospheric OCS budget and the role of OCS in climate.

**Acknowledgements**

The authors thank the Ocean Biology and Processing Group at NASA Goddard Space Flight Center for access to the Aqua MODIS data as well as ECMWF for access to the ERA5 data.

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

**Figures**

a)                                                            b)

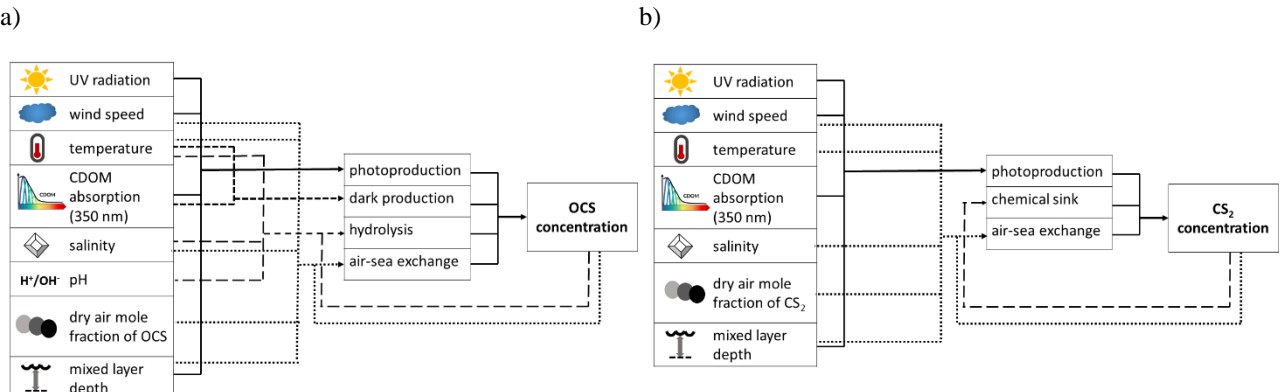

Figure 1: Schematic overview on processes and forcing included in the box models for a) OCS and b) CS$_2$.

a)

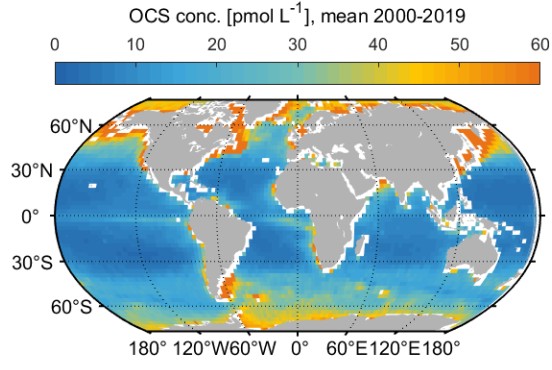

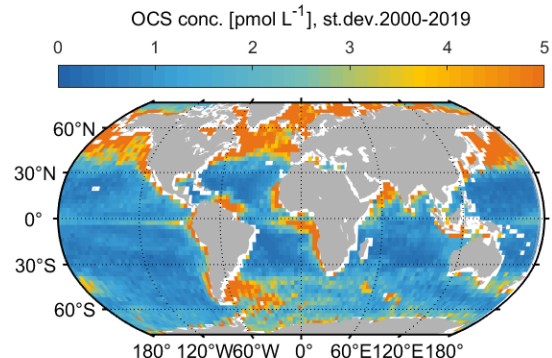

b)

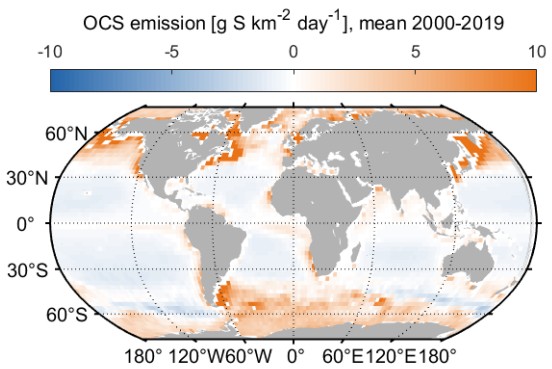

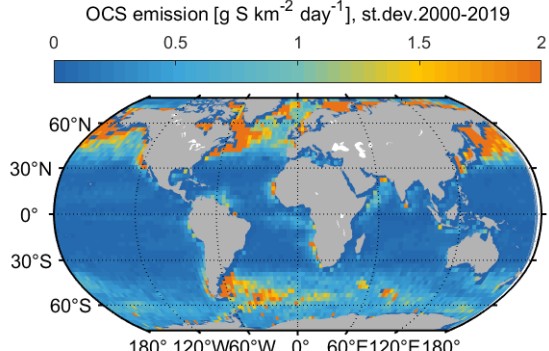

c)

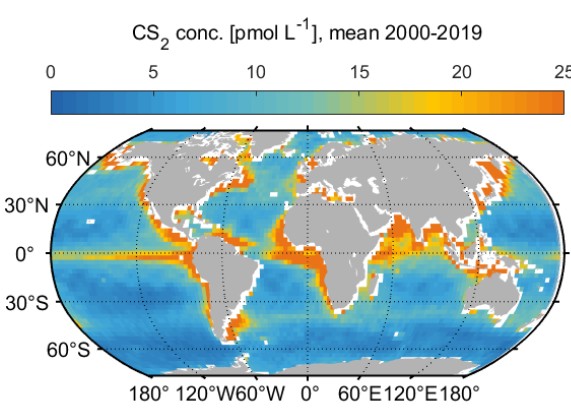

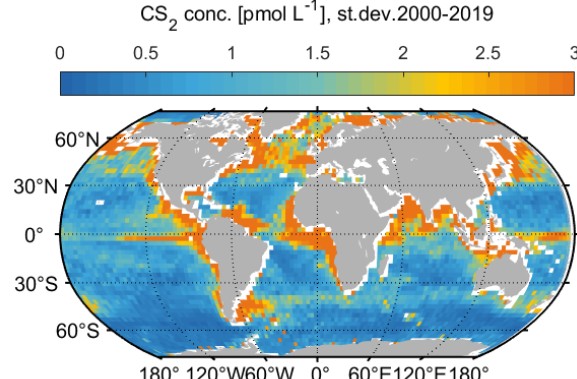

d)

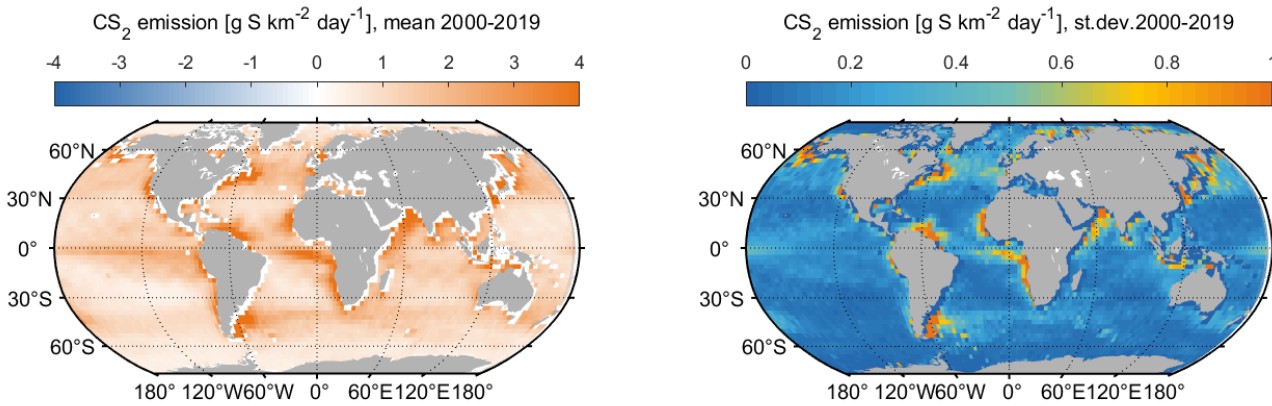

**Figure 2: Spatial variation of a) mean OCS surface concentration (left panel) and standard deviation of annual mean concentrations (right panel), b) same for OCS emissions, c) same for CS$_2$ surface concentration, d) same for CS$_2$ emissions, averaged over the 2000-2019 period.**


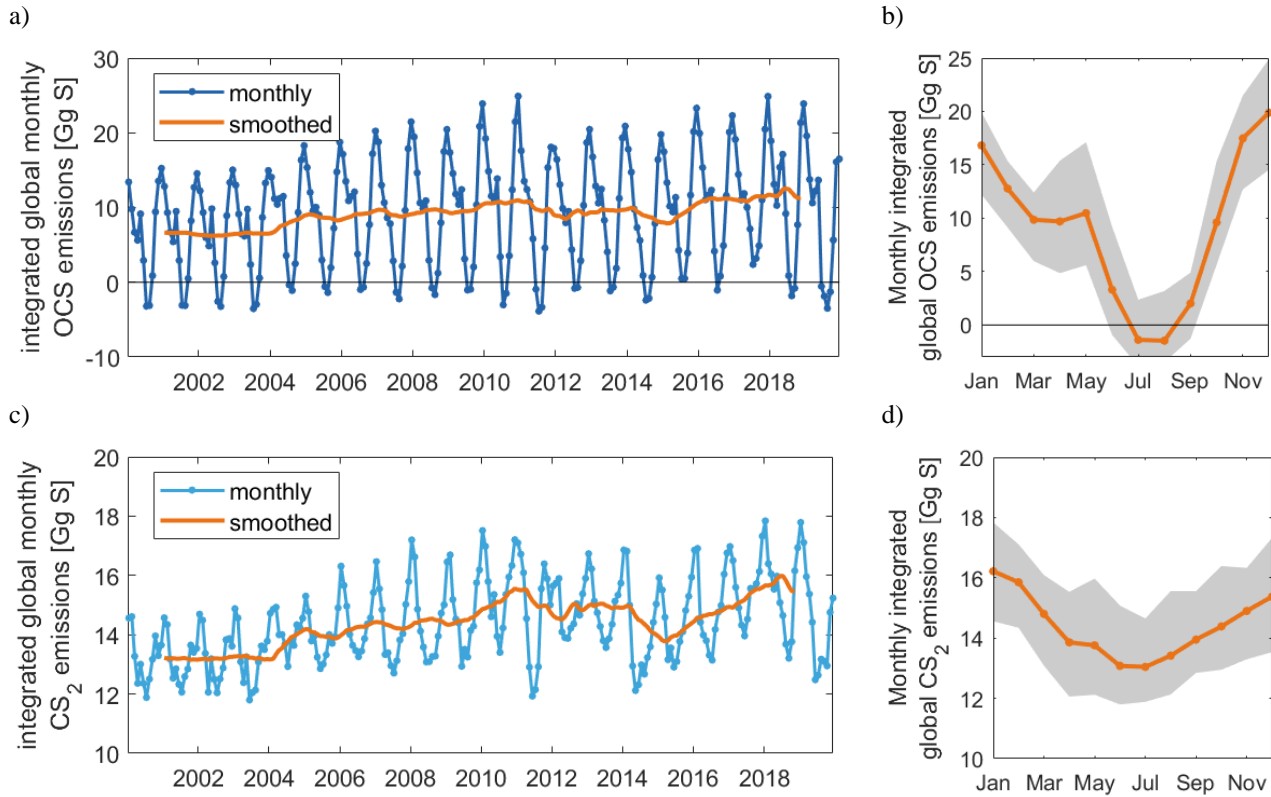

**Figure 3: Interannual variability of OCS emissions as time series (a) and mean annual cycle in orange, standard deviation of respective month in shaded grey area (b); (c) and (d) same as (a) and (b) but for CS$_2$. The model output is saved in 2-hour intervals for the 15$^{th}$ of each month, and integrated over 30 days for the monthly emissions shown here.**


**a)**

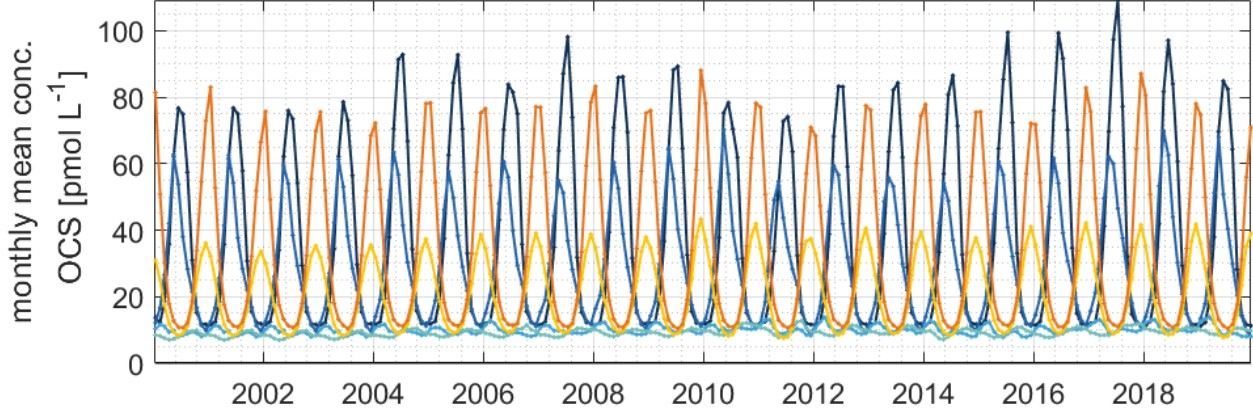

**b)**

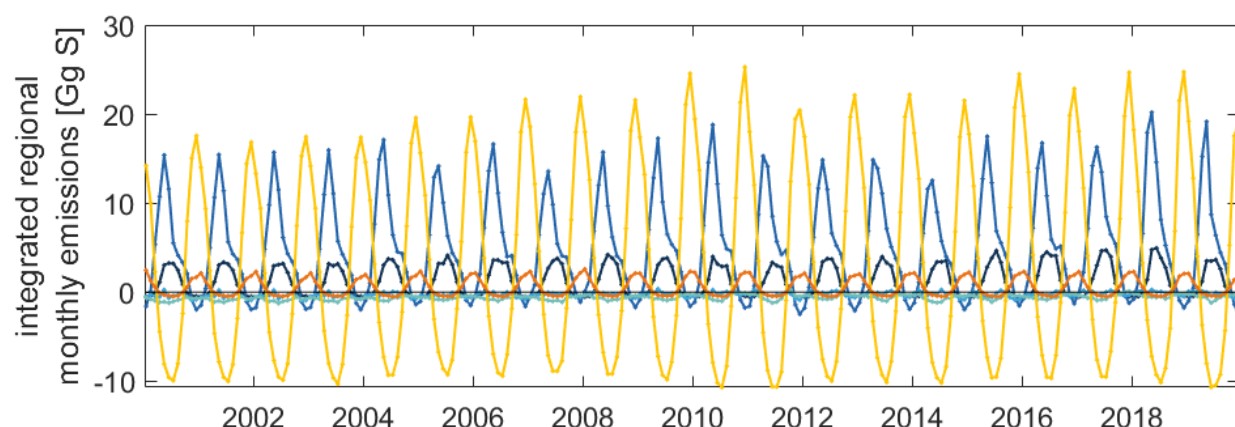

**c)**

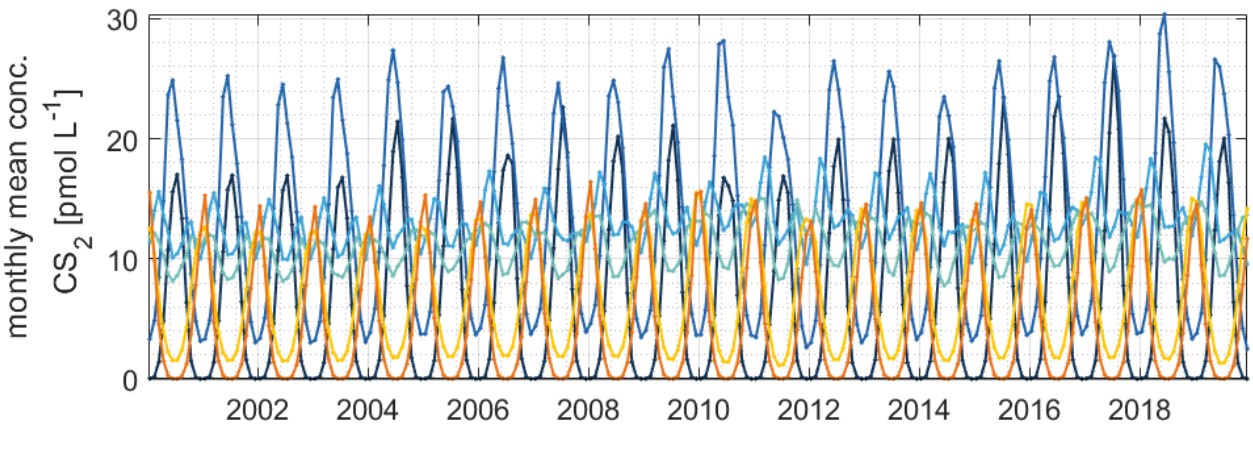

d)

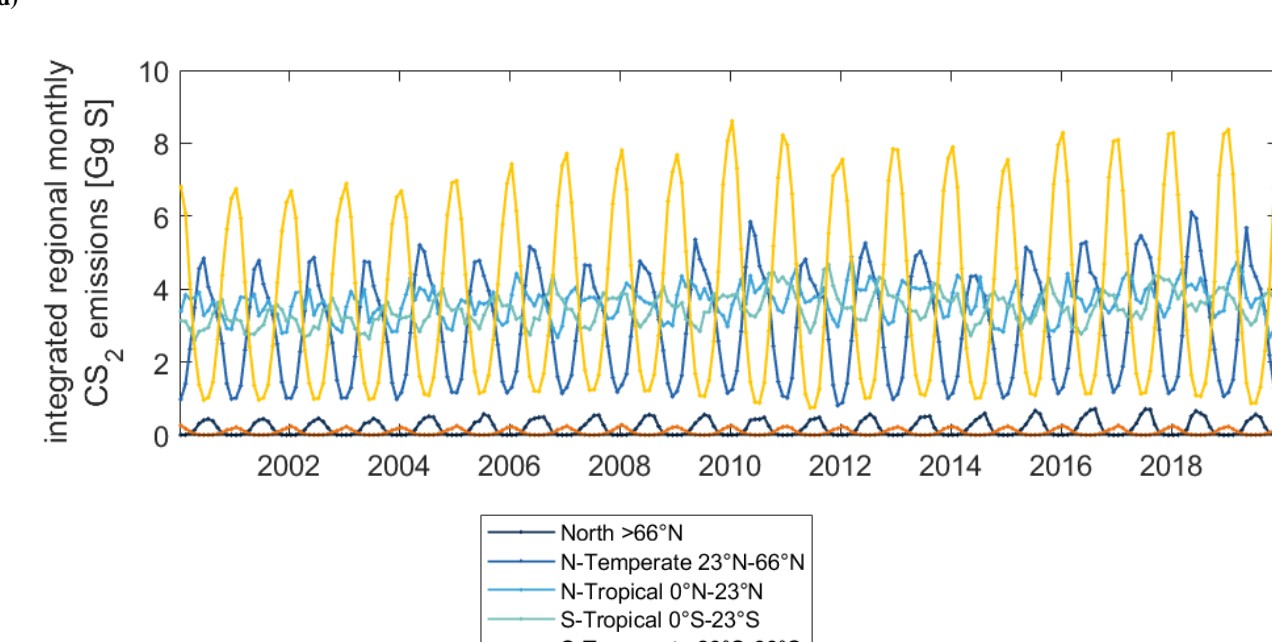

**Figure 4: Regionally resolved interannual variability of concentrations (a) and emissions (b) for OCS. Same in (c) and (d) for CS₂.**


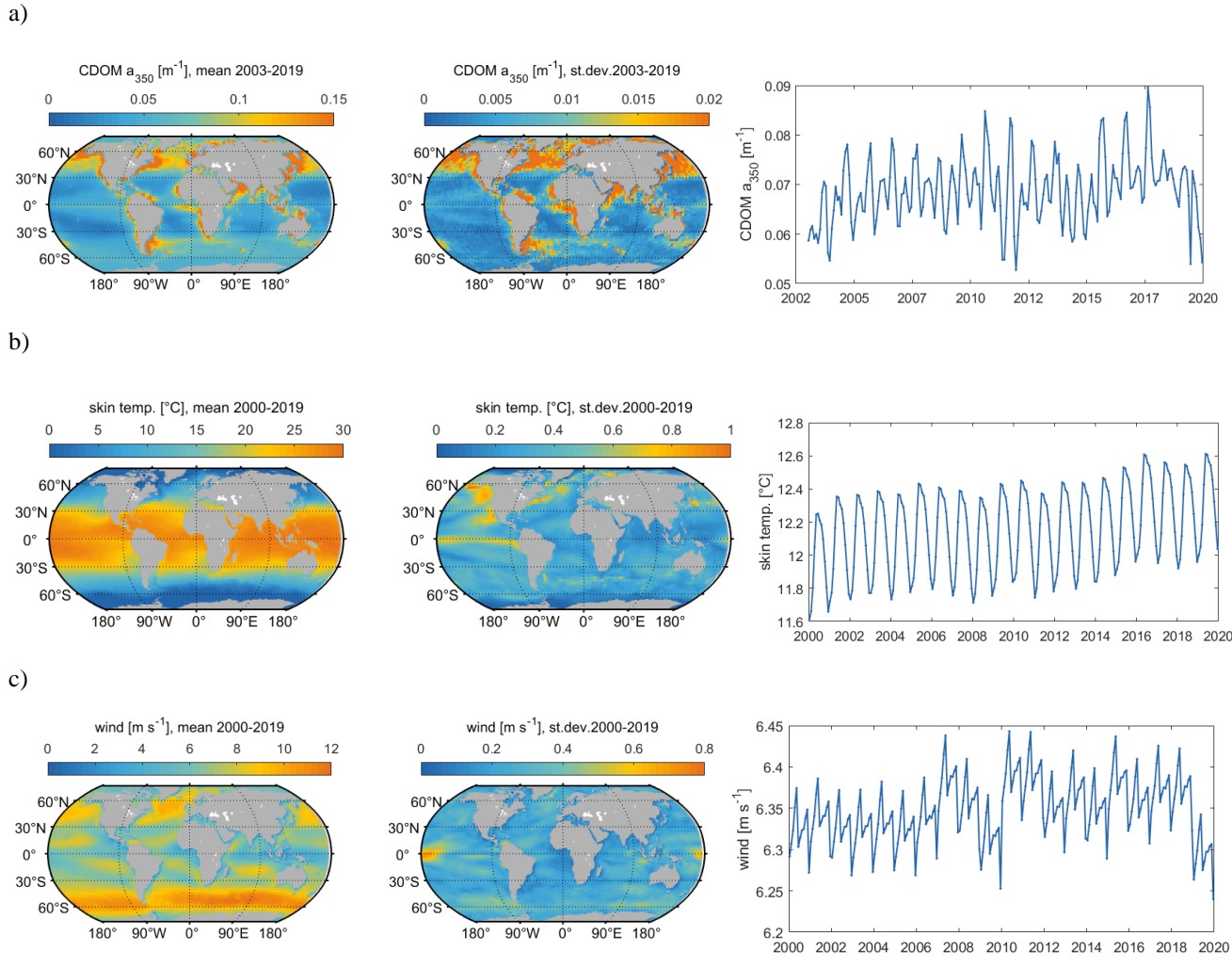

**Figure 5: Mean and standard deviation (maps) and interannual variation (right panels) of model input parameters: (a) CDOM a350, (b) skin temperature, (c) wind speed. Data sources listed in Tab. 1.**

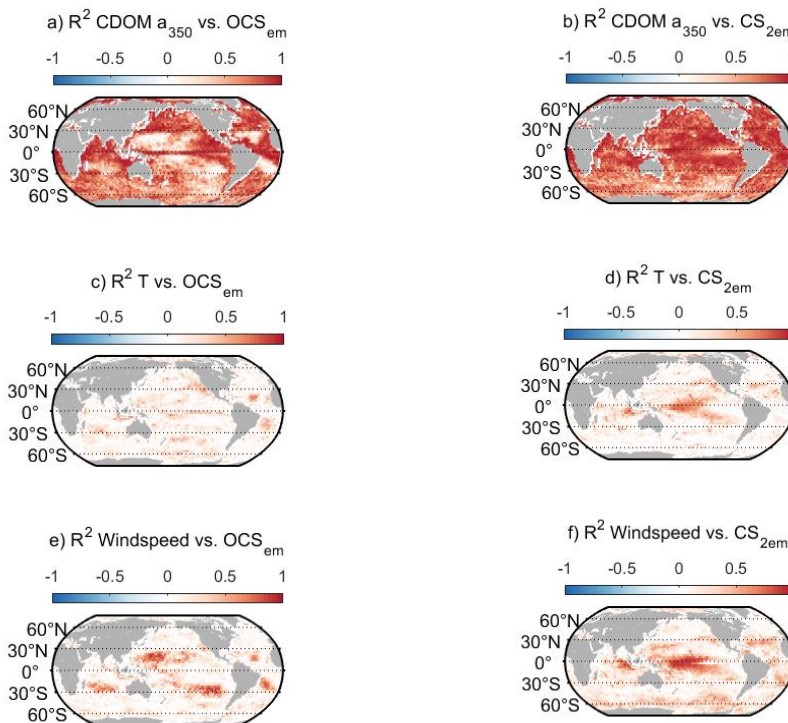

**Figure 6: Regional correlation of annual OCS (left column) and CS₂ (right column) emission data with monthly data for temperature (upper row), wind speed (middle row) and CDOM absorption coefficient (lower panel). Correlation is shown as Pearson's R².**


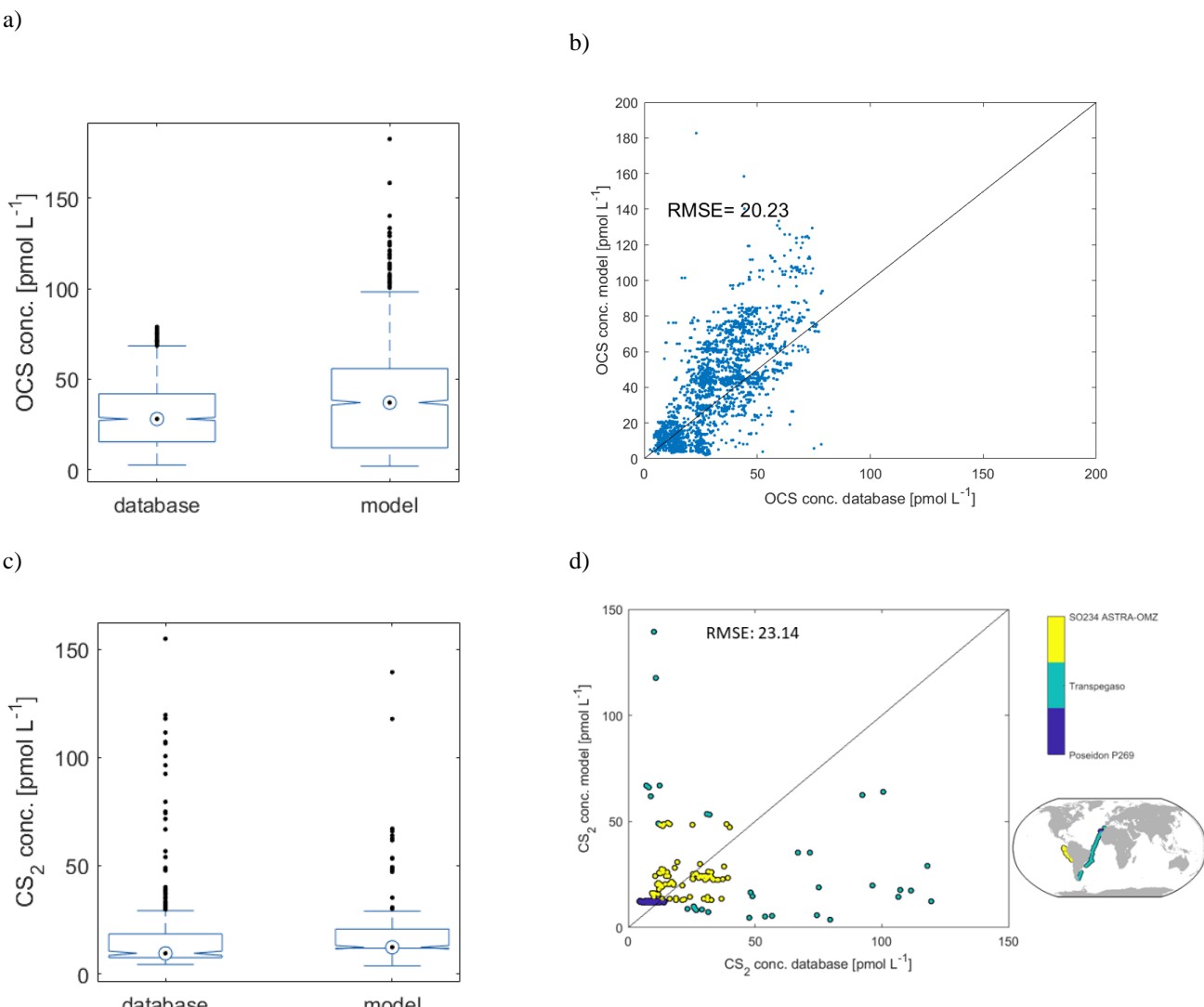

**Figure 7: Comparison of model-output to observations from the database described in Lennartz et al., (2020). a) Box plot of OCS reference data from database and subsampled model output at time and location of measurements (32 cruises) b) scatter plot of 1:1 comparison with same data as in a), black line is 1:1 line, c) and d) same as a) and b) but for CS₂ (3 cruises).**


**Tables**

Table 1: Overview on forcing parameters, their resolution and sources used for the box model simulations 2000-2019.

| Parameter | Resolution | Source |
|---|---|---|
| **Absorption coefficient of CDOM at 350 nm (a350)** | gridded, monthly resolution | Aqua MODIS satellite data, monthly composite of absorption due to gelbstof and detritus at 443, converted to 350 nm with a reference slope of 0.02. Note that years 2000-2002 are the same as 2003, as data is only available from late 2002 onwards. (NASA Goddard Space Flight Center, 2019) |
| **Surface (skin) temperature** | gridded, monthly resolution with mean diurnal cycle | ERA5 reanalysis (Hersbach et al., 2018), variable name in ERA5: 'skin temperature' |
| **Salinity** | gridded, climatological monthly mean | World Ocean Atlas 2013 (Levitus et al., 2013) |
| **Global radiation (converted to UV radiation)** | gridded, monthly resolution with mean diurnal cycle | ERA5 reanalysis (Hersbach et al., 2018), variable name in ERA5: 'surface solar radiation downwards' |
| **Wind speed at surface** | gridded, monthly resolution with mean diurnal cycle | ERA5 reanalysis (Hersbach et al., 2018), variable name in ERA5: 'u='10m u-component of wind' and v='10m v-component of wind' (for this study these were converted into total wind speed = sqrt($u^2+v^2$) ) |
| **pH** | constant value (8.1) | |
| **Mixed layer depth** | gridded, climatological monthly mean | MIMOC climatology (Schmidtko et al., 2013) |
| **Dry air mole fraction OCS** | constant value, 500 ppt | |
| **Dry air mole fraction CS$_2$** | constant value, 0 ppt | |
| **Sea surface pressure** | gridded, monthly resolution with mean diurnal cycle | ERA5 reanalysis (Hersbach et al., 2018), variable name in ERA5: 'surface pressure' |


**Table 2: Globally integrated annual emissions of OCS and CS2 for each year in 2000-2019, together with descriptive statistics and trends.**

| | OCS | CS$_2$ |
|---|---|---|
| | Gg S | Gg S |
| **2000*** | 81.3 | 160.8 |
| **2001*** | 77.3 | 160.0 |
| **2002*** | 78.0 | 161.2 |
| **2003** | 78.8 | 160.3 |
| **2004** | 108.3 | 172.0 |
| **2005** | 100.8 | 169.1 |
| **2006** | 116.3 | 175.3 |
| **2007** | 110.6 | 173.4 |
| **2008** | 114.4 | 175.0 |
| **2009** | 126.3 | 179.7 |
| **2010** | 133.3 | 189.2 |
| **2011** | 109.0 | 179.5 |
| **2012** | 113.3 | 181.2 |
| **2013** | 117.9 | 181.3 |
| **2014** | 97.2 | 170.1 |
| **2015** | 127.6 | 175.0 |
| **2016** | 134.7 | 181.5 |
| **2017** | 142.1 | 189.7 |
| **2018** | 136.9 | 187.8 |
| **2019** | 102.0 | 177.3 |
| mean | **110.3** | **174.97** |
| standard deviation | **20.3** | **9.3** |
| slope (only 2003-2019) | **1.7 Gg S yr$^{-1}$** | **0.95 Gg S yr$^{-1}$** |
| p slope (only 2003-2019) | **0.028** | **0.0067** |

*CDOM from 2003

**Table 3: Explained Variance (Pearson's R²) and significance level p for correlations of globally integrated emissions for OCS and CS₂ with global annual averages of CDOM a₃₅₀. skin temperature and wind speed. Significant results (α=0.01) are indicated in bold font.**

| | $CS_2$ | CDOM $a_{350}$ | temperature | wind |
|---|---|---|---|---|
| OCS | **$R^2$=0. 87** **p=2.1e-9** | **$R^2$=0.94** **p=1.0e-10** | **$R^2$=0.41** **p=0.0024** | **$R^2$=0.32** **p=0.0099** |
| $CS_2$ | 1 | **$R^2$=0.67** **p=6.0e-5** | **$R^2$=0.40** **p=0.0026** | $R^2$=0.29 p=0.0136 |
| CDOM $a$350 | | 1 | $R^2$=0.23 p=0.051 | $R^2$=0.22 p=0.0572 |
| temperature | | | 1 | $R^2$=0.02 p=0.53 |