# Peer review of "Monthly resolved modelled oceanic emissions of carbonyl sulfide and carbon disulfide for the period 2000-2019"

_Earth System Science Data, 2020_

## Referee Comment (RC1) · Anonymous Referee #1 · 7 Jan 2021

In general, this is a good and useful paper that provides a clear product for the modelling community: a monthly time series of OCS and CS2 fluxes from the ocean in the period 2000-2019. I have a number of suggestions that might improve the paper.

The basic formulas underpinning the model are presented in equations 1, 2, etc. I tried to grasp these formulas, but quite some details are missing. I am not suggesting to repeat the information from previous papers, but a full mention of units would be very helpful. For instance, equation 2: the non-trivial unit for the photochemical rate constant p (pmol per Joule, see Lennarz (2017)) has to be derived from the other units (which are given). I would be good to provide all units clear. Also, the link between the

main text and the figures and table could be improved (e.g. the assumed atmospheric mole fractions for OCS and CS2 are given only in a table).

The abstract could better reflect the method used in the paper, and should also mention the data that drive the model…something like, we use a 1D model of the ocean' mixed layer driven by ERA5 data from ECMWF and CDOM from MODIS…For "temperature" the ocean skin temperature is used. I agree with this choice, because it given information about the sea surface temperature. I still wonder, however, how sensitive the results are for alternative choices, such as Sea Surface Temperature from ERA (see e.g. Luo, B.; Minnett, P.J. Evaluation of the ERA5 Sea Surface Skin Temperature with Remotely-Sensed Shipborne Marine-Atmospheric Emitted Radiance Interferometer Data. Remote Sens. 2020, 12, 1873.).

I find the analysis of the "drivers" of flux variability in table 3 not very well described. I am a bit surprised that this analysis is performed on "global" and "yearly" data (monthly time-series of the global variables appear in figure 5). Although I clearly see that years with high CDOM are also years with higher emissions, I wonder if the global/yearly scale analysis if the most appropriate here. At least is should be made clear how the data are averaged (area-weighted?). But it might be more revealing to present a regional analysis.

Besides these points, the paper reads well, and provide interesting points of discussion. In reading the paper, I have annotated the pdf, which I include for further and minor remarks.

Please also note the supplement to this comment:
https://essd.copernicus.org/preprints/essd-2020-389/essd-2020-389-RC1-supplement.pdf

**Supplement:**

[revised manuscript text omitted]

---

## Referee Comment (RC2) · Anonymous Referee #2 · 22 Feb 2021

Monthly resolved modelled oceanic emissions of carbonyl sulfide and carbon disulfide for the period 2000–2019

This is a well written paper that provides up-to-date, gridded OCS and CS2 oceanic fluxes that have been evaluated against ocean surface layer observations. These fluxes are a badly needed update to products used in global models of OCS and will enable the community to better constrain the "missing" component of the OCS budget.

line 49: I would say multiple trace gases, rather than both. That would still be true. line 51: comparably? Technically you would have to say compared to what in the same sentence. You could say better understood than CS2. line 52: I had to admit that I

had to look up Diapycnal... line 64: I would say the mechanism is not well understood rather than well known.

What is the sign of dOCS_ase in Eqn 1? Should it be negative for release of OCS from the sea to the air?

line 168: Is there a reason you don't run the model at 0.25 x 0.25 degree instead of coarsening it to 2.8o? It would be good to add a sentence here justifying it.

line 175: How much uncertainty is introduced by using the 15th of the month instead of the monthly average conditions? I doubt it's too much but it would be good to have an idea of the magnitude mentioned here

line 205: I'd be careful to clarify that its a missing OCS source. Not CS2 as is implied with the wording here.

line 209: How realistic is that assumption (0ppt atmosphere)? And how dependent is the net source vs sink of this atmospheric conc? ie. How much CS2 would you need to have in the atmosphere to stop the net emission?

Fig 3: What is going on in 2019 to have so many negative OCS fluxes (Fig 4 suggests 60-90N might be driving this; Fig5 says combined CDOM and wind minimum)? And what happened in 2017 that there is year-round flux (seems like high Arctic is really high; CDOM also really high)? A sentence or two would be really interesting. I was expecting something different in 2015/2016 for the El Nino too. Could you mention why there is no significant change for that period?

line 290: CS2 emissions correlate with temperature even though no direct driver. If you look at a specific lat/time (ie. hold radiation $\sim$constant) does that correlation hold over the various years?

Section 4.4: What is the slope of the fit in Fig 6b? This is a really great comparison so I wonder if you could calculate a correction factor from the bias to apply based on the comparison with the data? Fig 6d: the Yellow dots are a little hard to see. Could you

add an outline to the symbols or something?

line 334: A sentence here explaining what the difference is between the cruises would be useful. Even just to call out the different areas sampled Fig 6d inset

line 348: I would say explicitly OCS and CS2 (instead of both) as its your last paragraph of discussion and people often flick through to check it out (like a conclusions section)

line 360. This is a great paper! Nice job. My one question at the end was: Did you find evidence for the large oceanic source that the top down studies were saying had to be there? Based on everything you presented here, the answer is no. And I think it's ok to say that! This is still a great product that the modelers will need to find the real source of the budget mismatch.

––––––––––––––––––––––––––

---

## Author Comment (AC1) · 19 Mar 2021

We thank the reviewers for their comments that helped us to improve the manuscript and make it more useful for the reader and users. Please find our replies below the respective comments (in bold font).

In general, this is a good and useful paper that provides a clear product for the modelling community: a monthly time series of OCS and CS2 fluxes from the ocean in the period 2000-2019. I have a number of suggestions that might improve the paper.

The basic formulas underpinning the model are presented in equations 1, 2, etc. I tried to grasp these formulas, but quite some details are missing. I am not suggesting to repeat the information from previous papers, but a full mention of units would be very helpful. For instance, equation 2: the non-trivial unit for the photochemical rate constant p (pmol per Joule, see Lennarz (2017)) has to be derived from the other units (which are given). I would be good to provide all units clear.

**As suggested, we have added the units to the equations.**

Also, the link between the main text and the figures and table could be improved (e.g. the assumed atmospheric mole fractions for OCS and CS2 are given only in a table).

**We agree and have added some additional descriptions to better link the content of the text and the tables/figures. In particular, we have added:**

**l. 104: "The numerical model simulating OCS seawater concentration and emission includes the processes photochemical production, light-independent production (termed 'dark production'), degradation by hydrolysis and air-sea exchange across the sea surface. The process rates are calculated as depicted in Fig. 1 based on meteorological (global radiation, wind speed, skin temperature) and physicochemical data (salinity, seawater pH, CDOM absorption, and dry mole air fraction). The processes photochemical production $\frac{d[OCS]_{photo}}{dt}$, dark production $\frac{d[OCS]_{dark}}{dt}$, hydrolysis $\frac{d[OCS]_{hydrolysis}}{dt}$ and air-sea exchange $\frac{d[OCS]_{ase}}{dt}$ are calculated according to equation (1), all in $\left[\frac{pmol}{L \cdot s}\right]$ (Fig. 1):"**

**l. 141: "…based on the atmospheric dry mole fraction".**

**In combination with a review comment by reviewer #2 we have added a more detailed discussion on the time series and linked it to the time series figures.**

The abstract could better reflect the method used in the paper, and should also mention the data that drive the model . . . something like, we use a 1D model of the ocean's mixed layer driven by ERA5 data from ECMWF and CDOM from MODIS.

**We agree and have changed the sentence to: "Emissions are calculated with a numerical box model (resolution 2.8° x 2.8° at equator, T42 grid) for the surface mixed layer, driven by ERA5 data from ECMWF and CDOM from Aqua-MODIS."**

For "temperature" the ocean skin temperature is used. I agree with this choice, because it given information about the sea surface temperature. I still wonder, however, how sensitive the results are for alternative choices, such as Sea Surface Temperature from ERA (see e.g. Luo, B.; Minnett, P.J. Evaluation of the ERA5 Sea Surface Skin Temperature with Remotely-Sensed Shipborne Marine-Atmospheric Emitted Radiance Interferometer Data. Remote Sens. 2020, 12, 1873.).

**Thanks for pointing us towards this issue. We have chosen skin temperature rather than sea surface temperature, because it is diagnosed closer to the air-sea interface where the exchange happens. As this choice may also have an effect on other temperature-relevant processes, we have**

performed new simulations using sea surface temperature as input. We have clarified this throughout the manuscript.

l. 167: Skin temperature is used as forcing data for all temperature-relevant processes, i.e. air-sea exchange, dark production and hydrolysis. To test the sensitivity of emissions on the choice between skin and sea surface temperature, we performed a sensitivity test for the year 2000.

l. 377: Another source of uncertainty is the forcing data, e.g. the choice of using the skin temperature rather than the sea surface data. For comparison, we performed a shorter simulation covering the year 2000 and using the ERA5 sea surface temperature data instead of the skin temperature. The difference in resulting global emissions was 1.2%, i.e. very small compared to other uncertainties.

l. 403: The calculated $CS_2$ emission estimate is not sensitive towards the choice of the temperature forcing data, resulting differences in global emissions when using the sea surface temperature instead of the skin temperature for the year 2000 resulted in a negligible deviation of 0.12%.

I find the analysis of the "drivers" of flux variability in table 3 not very well described. I am a bit surprised that this analysis is performed on "global" and "yearly" data (monthly time-series of the global variables appear in figure 5). Although I clearly see that years with high CDOM are also years with higher emissions, I wonder if the global/yearly scale analysis if the most appropriate here. At least is should be made clear how the data are averaged (area-weighted?). But it might be more revealing to present a regional analysis.

We agree with the reviewer that a regional analysis provides additional useful information, and have added a paragraph on this issue. We have also clarified in the main text how the data was averaged, i.e. an area-weighted average was used (we added a clarification in the text).

We have added a new figure (Figure 6), showing the correlation coefficients for annual OCS emissions and forcing data (CDOM, temperature and wind speed) for each grid point, and added the following text passages:

l. 320: Resolving the correlations regionally shows distinct controls on interannual variability for CDOM and wind speed, but not for temperature (Fig. 6). Highest Pearson's correlation coefficients ($R^2$) for CDOM and OCS emission are found globally except for the subtropical gyres (Fig. 6a). In those gyre regions, CDOM concentration is generally low (Fig. 5a), so that other drivers like wind speed seem to have a higher impact on the variability (Fig. 6e). Correlations with temperature show no clear spatial pattern (Fig. 6c).

l. 336: Regional analysis of correlations of $CS_2$ emissions with biogeochemical and meteorological data shows that CDOM is a globally homogeneous driver of emissions as indicated by the high Pearson's correlation coefficients globally. Temperature and wind speed show highest correlation to $CS_2$ emissions in the tropical West Pacific, where the assumed source region of the 'missing source' of OCS is located. In these regions, interannual variability of wind speed is highest (Fig 5), and temperature shows increased variability there (Fig. 5). This increased variability might explain the regionally strong correlation with $CS_2$ emissions.

Note that we have also performed the regional analysis with the monthly data but obtained less clear results. Given the strong annual cycles that we show in Figure 3 of the manuscript, much of the monthly variability is explained by seasonal patterns. We thus decided not to present this data in the main manuscript, as it does not provide much additional information.

Besides these points, the paper reads well, and provide interesting points of discussion. In reading the paper, I have annotated the pdf, which I include for further and minor remarks.

**We have corrected the typos and addressed the remarks in the annotated version.**

**Reviewer #2**

Monthly resolved modelled oceanic emissions of carbonyl sulfide and carbon disulfide for the period 2000–2019

This is a well written paper that provides up-to-date, gridded OCS and CS2 oceanic fluxes that have been evaluated against ocean surface layer observations. These fluxes are a badly needed update to products used in global models of OCS and will enable the community to better constrain the "missing" component of the OCS budget.

line 49: I would say multiple trace gases, rather than both. That would still be true.

**Changed as suggested.**

line51: comparably? Technically you would have to say compared to what in the same sentence. You could say better understood than CS2.

**Changed as suggested.**

line 52: I had to admit that I had to look up Diapycnal...

**Thanks for this comment to help making this accessible to an interdisciplinary community – we have added an explanation.**

**"Gas fluxes across the base of the mixed layer, i.e. diapycnal fluxes, seem to be of minor importance, …"**

line 64: I would say the mechanism is not well understood rather than well known.

**Changed as suggested.**

What is the sign of dOCS_ase in Eqn 1? Should it be negative for release of OCS from the sea to the air?

**Thanks for spotting that mistake – we corrected it! It should indeed be negative, as emissions to the atmosphere are a sink to OCS in the surface ocean.**

line 168: Is there a reason you don't run the model at 0.25 x 0.25 degree instead of coarsening it to 2.8? It would be good to add a sentence here justifying it.

**The resolution is the same as in Lennartz et al., 2017, suitable as an input for atmospheric models such as the EMAC atmospheric chemistry climate model. More importantly, given the large uncertainties carried in the global parameterization of the photochemical production or the air-sea exchange, a finer resolution would suggest a higher level of certainty than there actually is. We have added the following:**

**"The spatial resolution is the same as in Lennartz et al., (2017)."**

line 175: How much uncertainty is introduced by using the 15th of the month instead of the monthly average conditions? I doubt it's too much but it would be good to have an idea of the magnitude mentioned here

**This seems to be a misunderstanding: We use a monthly mean average (not the data of the 15th), but apply it as the diel cycle of the 15th of each month. Then, we interpolate between the diel cycles of each month's 15th. This ensures that we do not have sharp changes as if we would use the mean monthly cycle for each day of the month. In order to clarify this, we have added/changed the following:**

**"The average diel cycle of each meteorological dataset (wind, pressure, skin temperature and solar radiation) is used for the 15th of each month (one value for every 2 hours). In between, data is interpolated separately for each time of the day, resulting in a continuous change of the amplitude of the diel cycles. This procedure avoids sharp changes as if a mean monthly cycle was used for each day of the months, while still being computationally effective."**

line 205: I'd be careful to clarify that it's a missing OCS source. Not CS2 as is implied with the wording here.

**We have changed the sentence so that it becomes clear that it is the "missing source of OCS,…".**

line 209: How realistic is that assumption (0ppt atmosphere)? And how dependent is the net source vs sink of this atmospheric conc? ie. How much CS2 would you need to have in the atmosphere to stop the net emission?

**We agree that it is difficult to put a number on the $CS_2$ concentration in the remote marine boundary layer that is representative, given the small number of observations with most of them close to land. We have computed the influence of air mixing ratios with an average temperature of 20°C, a salinity of 34.5 psu, and a wind speed of 5 m/s, and found differences of up to 30% between 0 and 40 ppt. We have added the following:**

**"This assumption is a simplification, the average of the sparse (n=901) dataset on available $CS_2$ air mixing ratios is 42±24 ppt, but ranging to not detectable in remote ocean regions. The difference can be up to 30% in the computed flux, similar to the uncertainty inherent to the computation of the transfer velocity."**

Fig 3: What is going on in 2019 to have so many negative OCS fluxes (Fig 4 suggests 60-90N might be driving this; Fig5 says combined CDOM and wind minimum)? And what happened in 2017 that there is year-round flux (seems like high Arctic is really high; CDOM also really high)? A sentence or two would be really interesting. I was expecting something different in 2015/2016 for the El Nino too. Could you mention why there is no significant change for that period?

**We added a brief discussion of the specific points in the time series as you suggested in the section where we discuss the drivers of interannual variability:**

**l. 293: The variability comprises years like 2015 or 2017, in which positive OCS emissions occur in every month of the year, and years like 2019, where global net uptake by the ocean was present in four of the twelve months (Fig. 3a). Most of the interannual variability of these emissions are driven by the emissions in the high latitudes. For example, in 2017, emissions in the Arctic regions are higher than on average, and lead to an overall increase in the emissions even in the winter months. 2015/2016 was a strong El Nino year, and decreased upwelling of cold water with high CDOM content would expectably lead to low OCS emissions due to decreased photochemical and dark production, and increased hydrolysis due to warmer water temperatures. However, as fluxes in the tropics are generally small, the global emissions are note substantially lower compared to other years (for 2015 even higher, due to higher emissions in high latitudes). The many negative fluxes in 2019 seem to result from lower than average emissions in the Southern Ocean.**

line 290: CS2 emissions correlate with temperature even though no direct driver. If you look at a specific lat/time (ie. hold radiation~constant) does that correlation hold over the various years?

**In agreement with the comments of reviewer #1, we have extended the discussion about the interannual variability by adding a regional analysis. This analysis includes a new figure where we do the correlation for each grid point separately (for both gases).**

**We have added the following (in addition to a new figure 6):**
**l. 320: Resolving the correlations regionally shows regionally distinct controls on interannual variability for CDOM and wind speed, but not for temperature (Fig. 6). Highest Pearson's correlation coefficients ($R^2$) for CDOM  and OCS emission are found globally except for the**

**subtropical gyres (Fig. 6a). In those gyre regions, CDOM concentration is generally low (Fig. 5a), so that other drivers like wind speed seem to have a higher impact on the variability (Fig. 6e). Correlations with temperature show no clear spatial pattern (Fig. 6c).**

**l. 336: Regional analysis of correlations of $CS_2$ emissions with biogeochemical and meteorological data shows that CDOM is a globally homogeneous driver of emissions as indicated by the high Pearson's correlation coefficients globally. Temperature and wind speed show highest correlation to $CS_2$ emissions in the tropical West Pacific, where the assumed source region of the 'missing source' of OCS is located. In these regions, interannual variability of wind speed is highest (Fig 5), and also temperature shows increased variability there (Fig. 5). This increased variability might explain the regionally strong correlation with $CS_2$ emissions.**

Section 4.4: What is the slope of the fit in Fig 6b? This is a really great comparison so I wonder if you could calculate a correction factor from the bias to apply based on the comparison with the data?

**We did calculate the slope and inverted the function to provide a way to correct modelled values. We have added the following:**

**A correction for this bias was obtained from a linear fit through the 1:1 comparison (blue dots in Fig 7), and yielded the equation [OCS corrected] = 0.83 x [OCS modelled] – 0.7. Because the bias is still within the scatter of the data, we did not apply this correction factor for the analysis presented here.**

Fig 6d: the Yellow dots are a little hard to see. Could you add an outline to the symbols or something?

**We added black outlines to the symbols as suggested.**

line 334: A sentence here explaining what the difference is between the cruises would be useful. Even just to call out the different areas sampled Fig 6d inset.

**We agree and have added:**
**The three cruises cover the Mauretanean upwelling (Poseidon 269, blue in Fig. 7d), the Peruvian upwelling (ASTRA-OMZ, yellow in Fig. 7d) and a transect through the Atlantic (Transpegaso, green in Fig. 7d). As such, they cover a broad range of different biogeochemical regimes, but regions such as oligotrophic gyres or high latitude waters are not covered, i.e. a substantial part of the global variability might be missing in the reference dataset.**

line 348: I would say explicitly OCS and CS2 (instead of both) as its your last paragraph of discussion and people often flick through to check it out (like a conclusions section)

**Thanks, we agree and changed as suggested.**

line 360. This is a great paper! Nice job. My one question at the end was: Did you find evidence for the large oceanic source that the top down studies were saying had to be there? Based on everything you presented here, the answer is no. And I think it's ok to say that! This is still a great product that the modelers will need to find the real source of the budget mismatch.

**Thank you! We have been reluctant to express this conclusion in a data description paper, but we agree that it is a logical question one might ask after reading this paper. We have thus added a sentence in the conclusion section:**

**l. 435: Based on the data presented here, it seems unlikely that the missing atmospheric source of 600-800 Gg S $yr^{-1}$ (Berry et al., 2013; Glatthor et al., 2015; Kuai et al., 2015a) might be balanced by tropical marine emissions of OCS or $CS_2$.**